# Light-Induced Self-Oscillations and Spoiling of the Bragg Resonance Due to Nonlinear Optical Propagation in Heliconical Cholesteric Liquid Crystals

**Ashot H. Gevorgyan** [1] and **Francesco Simoni** [2,3,*]

[1] School of Natural Sciences, Far Eastern Federal University, 10 Ajax Bay, Russky Island, 690922 Vladivostok, Russia

[2] Università Politecnica delle Marche, 60100 Ancona, Italy

[3] Institute of Applied Sciences and Intelligent Systems CNR, 80072 Pozzuoli, Italy

[*] Correspondence: f.simoni@photomat.it

**Abstract:** In a recent paper, we have reported the results of a study of the nonlinear light propagation of a beam traveling along the helix direction of a heliconical cholesteric liquid crystal, showing that optical reorientation leads to instabilities in the optical transmission when the light wavelength is close to the Bragg resonance. Here we report a detailed study of this phenomenon, using Ambartsumian's layer addition modified method to take into account the continuous modification of the wave field during propagation. We show that the whole transmission spectrum is modified by increasing the light intensity and point out that self-induced oscillations take place at lower intensities on the red side edge of the Bragg resonance while stable values of transmittivity are still observed on the blue side edge. A further increase in the intensity leads to oscillations of lower amplitude on the blue side while an irregular behavior of the transmission is achieved on the red side. At higher intensities, the Bragg resonance disappears and transmission becomes unstable for any light wavelength. A simple phenomenological model is proposed to account for the onset of the oscillations and the asymmetry of the behavior at the opposite side of the Bragg resonance. We also point out that the static electric field is a driving parameter to switch from stable to oscillatory to irregular behavior in the transmittivity at a given light wavelength.

**Keywords:** heliconical liquid crystals; nonlinear optics; pitch tuning; optical instabilities

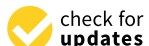



## 1. Introduction

It has been recently highlighted that heliconical cholesteric liquid crystals (ChOH, oblique helicoidal cholesterics) offer the opportunity to observe light-induced effects that are not possible in conventional planar cholesterics (CLC). It has been pointed out that the new phenomenology linked to the conical arrangement of the molecular director **n** originated from a bend elastic constant $K_3$ lower than the twist elastic constant $K_2$ ($K_3 < K_2$) and requires the application of a low-frequency (static) electric field $E_S$ in a direction parallel to the helix axis and within a range of values between two critical fields $E_{N^*C} < E_S < E_{NC}$. If $E_S < E_{N^*C}$ has a conventional CLC structure with a helix axis rotated by 90° with respect to the conical configuration, for $E_S > E_{NC}$ a complete unwinding of the structure occurs [1–5].

Unprecedented tuning of the Bragg resonance over the whole visible spectrum is possible by scanning a low-frequency electric field [1–3] over fractions of V/mm, providing possible easy applications to tunable optical filtering [6]. Additionally, light-induced optical reorientation has been demonstrated, leading to the tuning of the Bragg photonic band gap (PBG) and control of the light polarization state from linear to elliptical to circular in dependence on light intensity [7,8]. It has been already underlined that effective optical reorientation due to the torque action of the optical field $E_{OPT}$ on the molecular director **n**

is possible in ChOH because of the bend deformation of the conical structure, which is not present in conventional CLCs [7], where optical reorientation would necessarily involve a twist deformation, requiring field values higher by a few orders of magnitude to occur.

　　The effects of such optical reorientation on the nonlinear propagation of a light-beam-inducing optical reorientation have also been theoretically investigated [9]. In fact, light propagation through this structure may have a strong nonlinear character due to the bend deformation and the consequent optical torque originated by the interaction between the optical field and the molecular director **n**. This torque competes with the one originated by the static field; the result is an increase in the conical angle $\theta$ and, as a consequence, of the helix pitch. The geometry of the interaction and competing actions of the static and optical fields is sketched in Figure 1. This leads to a redshift of the Bragg resonance, producing a nonlinear transmission of the light beam strongly dependent on its wavelength as it approaches the PBG.

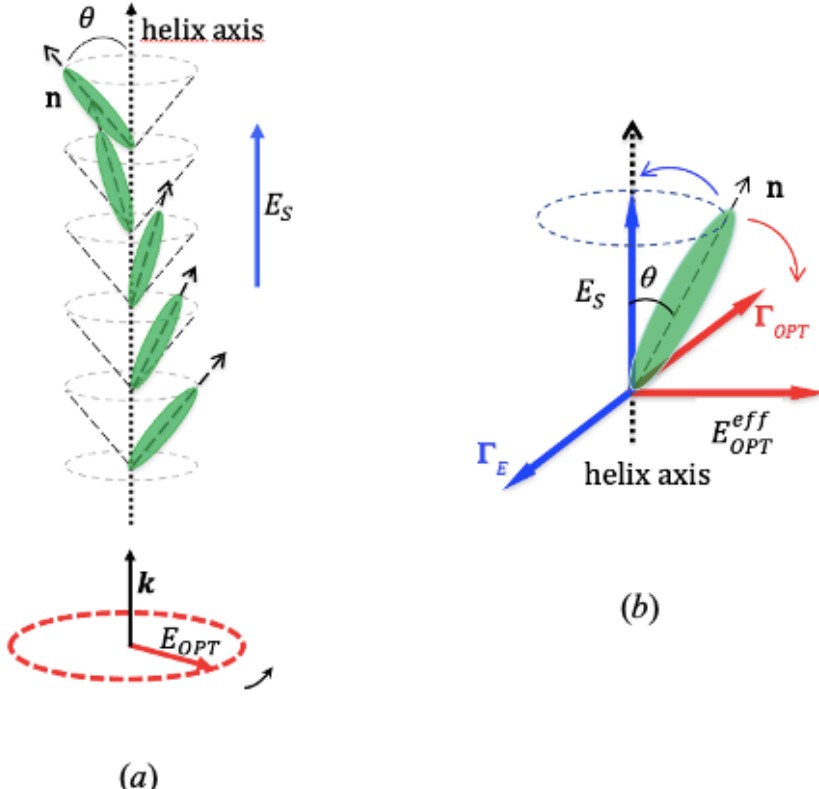

**Figure 1.** The analyzed geometrical configuration. (**a**) The optical field $E_{OPT}$ propagates with wavevector $k$ along the direction of the helix axis of the heliconical structure being circularly polarized in the transversal plane; the static field $E_S$ is applied along the direction of the helix axis. (**b**) The competition of the optical torque $\Gamma_{OPT}$ and the static field torque $\Gamma_E$ acting on the molecular director toward the opposite direction. $E_{OPT}^{eff}$ represents the component of the optical field effective at a given time.

　　This is easily expected if we extend to ChOH the theory of light propagation along the helical axis of CLC using the following substitution in all the relevant mathematical expressions [9]:

$$\epsilon_{\parallel} \;\rightarrow\; \epsilon_{eff} = \frac{\epsilon_{\parallel}\epsilon_{\perp}}{\epsilon_{\parallel} - \left(\epsilon_{\parallel} - \epsilon_{\perp}\right)\sin^2\theta} \tag{1}$$

In this way, the standard expression for the reflectance of the resonant mode can be used [10]:

$$R = \frac{\gamma^2 \sin h^2(sd)}{s^2 \cos h^2(sd) + \left(\frac{\Delta k}{2}\right)^2 \sin h^2(sd)} \tag{2}$$

where:

$$\gamma = \frac{\pi \Delta \epsilon_{eff}}{2\lambda_0 \sqrt{\tilde{\epsilon}}} \; ; \; s = \sqrt{\gamma^2 - \left(\frac{\Delta k}{2}\right)^2} \; ; \; \frac{\Delta k}{2} = k - q \tag{3}$$

using the definitions given in [9] here and in the following.

Here $k$ is the resonant mode wavevector, which for a left-handed helix is:

$$k_L = \tilde{\epsilon}^{1/2} k_0 - \frac{\alpha^2 k_0^3}{8q\tilde{\epsilon}^{\frac{1}{2}}\left(\tilde{\epsilon}^{\frac{1}{2}} k_0 - q\right)} \tag{4}$$

Additionally, it is necessary to take into account the dependence of the conical angle and the pitch on the effective field:

$$E_{eff} = \sqrt{E_S^2 - \frac{\Delta \epsilon_{OPT}}{\Delta \epsilon} \frac{4\pi}{n_{av}c} I} \tag{5}$$

$$\sin^2 \theta = \frac{\kappa}{1-\kappa}\left(\frac{E_{NC}}{E_{eff}} - 1\right) \quad \text{and} \quad p = \kappa \frac{E_{NC} p_0}{E_{eff}} \tag{6}$$

This can be done analytically by considering a constant value of the intensity over the whole sample thickness. At a fixed value of the applied static field in the range of stability of the heliconical structure, the intensity dependence of light transmission spectra $T(I) = 1 - R(I)$ can be plotted by using Equation (2) and shows a strong light-induced redshift of the whole spectrum [9].

The constant intensity approximation (corresponding to a uniform sample layer) is certainly well fulfilled in a pump–probe configuration when the wavelength of the light beam inducing the optical reorientation is far from the Bragg resonance and the probe beam near the Bragg resonance is weak enough to induce a negligible interaction with the molecular director. On the other hand, in a single-beam configuration where optical reorientation is induced by the same beam that is traveling through the medium with a wavelength close to the Bragg resonance, this strong modulation of transmission may affect the light propagation in a more complex way if the change in intensity during propagation cannot be neglected. To take into account this effect, a more careful calculation has been recently carried out based on Ambartsumian's layer addition modified method [11]. This method allows calculating the components of fields reflected and transmitted by the sample and also the field inside the material, making it possible to study the features of the localization of light in the system. Under this approach, the sample is considered a stratified material, and the optical field is recalculated after traveling through each sublayer; in this way, the effects of the light field are determined more carefully. These calculations have shown that, over a typical intensity dependence on the material parameters, instabilities occur in the optical transmission as a result of the nonlinear light propagation.

We present here a detailed analysis of this phenomenon, pointing out that, over a threshold intensity, stable oscillations take place until at higher intensities a chaotic behavior is reached. We demonstrate the strong asymmetric behavior in the spectral regions around the Bragg peak, where an oscillatory behavior at longer wavelengths corresponds to a uniform transmission at shorter ones. The possibility of switching from stable to oscillatory behavior by the static electric field is also shown. The phenomenon originates from the light-induced shift of the Bragg resonance, which affects light transmission during propagation, as will be described below.

## 2. Method and Results

We start from the conventional dielectric permittivity and magnetic permeability tensors of a ChOH:

$$\hat{\varepsilon}(z) = \begin{pmatrix} \widetilde{\epsilon} + \frac{\Delta\epsilon_{eff}}{2}\cos(2qz) & \frac{\Delta\epsilon_{eff}}{2}\sin(2qz) \\ \frac{\Delta\epsilon_{eff}}{2}\sin(2qz) & \widetilde{\epsilon} - \frac{\Delta\epsilon_{eff}}{2}\cos(2qz) \end{pmatrix}, \quad \hat{\mu}(z) = \hat{I} \tag{7}$$

We remind that $\Delta\epsilon_{eff} = \epsilon_{eff} - \epsilon_{\perp}$, $\Delta\epsilon_{OPT} = \epsilon_{\parallel} - \epsilon_{\perp}$, where $\epsilon_{\parallel}$ and $\epsilon_{\perp}$ are the dielectric permittivity parallel and perpendicular to the director in the nematic phase. Dividing the sample thickness into a large number of sublayers small enough to consider as constant the optical parameters of each sublayer, the problem consists of the solution of the following system of matrix difference equations:

$$\hat{R}_m = \hat{r}_m + \widetilde{\hat{t}}_m \hat{R}_{m-1}\left(\hat{I} - \widetilde{\hat{r}}_m \hat{R}_{m-1}\right)^{-1}\hat{t}_m,$$
$$\hat{T}_m = \hat{T}_{m-1}\left(\hat{I} - \widetilde{\hat{r}}_m \hat{R}_{m-1}\right)^{-1}\hat{t}_m, \tag{8}$$

with initial conditions $\hat{R}_0 = \hat{0}$ and $\hat{T}_0 = \hat{I}$ [12,13].

Here $\hat{R}_m$, $\hat{T}_m$, $\hat{R}_{m-1}$, and $\hat{T}_{m-1}$ are the reflectance and transmittance matrices of the system with $m$ and $(m-1)$ sublayers, respectively; $\hat{r}_m$, $\hat{t}_m$ are the reflectance and transmittance matrices of the $m$-th sublayer; $\hat{0}$ is the zero matrix; $\hat{I}$ is the unit matrix; the respective matrices for the reverse light propagation are denoted by a tilde. The method allows us to find the reflection coefficient $R = \frac{|E_r|^2}{|E_i|^2}$, the transmission coefficient $T = \frac{|E_t|^2}{|E_i|^2}$, and light intensity of the wave traveling in the ChOH sample $I(z) = |E_{in}(z)|^2 I_i$. We consider the eigenmode with circular polarization with the same handedness as the ChOH helix, which is suffering Bragg diffraction. As usual, $E_i$, $E_r$, and $E_t$ are the fields of incident, reflected, and transmitted waves, respectively, and $I_i$ is the intensity of incident light. Taking $I_i = N I_0$, we operate with the normalized intensity $N$ considering in the first step ($j = 1$) of the calculation:

$$E_{eff} = \sqrt{E_S^2 - \frac{\Delta\epsilon_{OPT}}{\Delta\epsilon}N}, \quad p = \text{const and } s = \sin^2\theta = \text{const} \tag{9}$$

With these parameters, we calculated the reflection, transmission coefficients as well as field intensity in each location of the liquid crystal layer following the method presented in [12].

In this way, we carry out the calculation that allows obtaining $I(z, \lambda) = |E_{in}(z, \lambda)|^2 I_i$. In the second step ($j = 2$), we have the new effective field $E_{eff}(z, \lambda) = \sqrt{E_S^2 - \frac{\Delta\epsilon_{OPT}}{\Delta\epsilon}|E_{in}(z, \lambda)|^2 N}$ allowing us to take into account the distribution of the angle $\theta$ and the helix pitch along the sample as determined by the local effective field. Then we have $s(z, \lambda, j) = \sin^2\theta = \frac{\kappa}{1-\kappa}\left(\frac{E_{NC}}{E_{eff(z,\lambda,j)}} - 1\right)$ and $p(z, \lambda, j) = \kappa\frac{E_{NC}p_0}{E_{eff(z,\lambda,j)}}$. Again we get the reflection and transmission coefficients as well as the intensity $I(z, \lambda, j) = |E_{in}(z, \lambda, j)|^2 N$ for this new parameter, and so on by increasing the number of steps: $j = 3, 4, 5, \ldots$.

The material parameters chosen for the calculation are the ones reported in Ref. [11]: $\epsilon_{\parallel} = 2.79$, $\epsilon_{\perp} = 2.19$, $\Delta\epsilon = 4.79$, and $\kappa = 0.1$; the unperturbed helix pitch is $p_0 = 1400$ nm; the layer thickness is $d = 20$ μm with critical fields $E_{NC} = 4.88$ V/μm and $E_{N^*C} = 1.53$ V/μm; and we take $n_{av} = \sqrt{(\epsilon_{\parallel} + \epsilon_{\perp})/2}$, where $n_{av}$ is the refractive index of the ChOH layer. The static field applied along the helix direction is $E_S = 2.03$ V/μm. In our calculations all the above-mentioned sublayers have the same thickness: $d_1 = d_2 = d_3 = \ldots = d_L = 8$ nm.

We remark that computer programs were compiled in Visual Basic (based on analytical expressions for the reflection and transmission matrices for an anisotropic layer [13] and

the layer addition method [12]) and debugged by the authors, and the graphs were built using the Excel and MATLAB programs.

The basic results of our calculations are shown in Figure 2. Spectra of the light transmittivity of the ChOH sample are reported vs. increasing values of the step index $j$ for increasing values of the normalized beam intensity $N$. Since the increase in the step index corresponds to the adjustment of the material parameters to the updated value of the local light field, it is strictly correlated to the response time of the medium $\tau_{on}$; therefore, we should consider $j$ proportional to the elapsed time. In this way, these plots give us the evolution in time of the transmission spectra. First of all, we remark that at $N = 0$ the transmission spectrum has the well-known form shown in Figure 2a and the change of $j$ has no influence on this spectrum, for this reason only the spectrum at $j = 1$ is reported. In Figure 2b, data for $N = 1$ to $N = 400$ are displayed. At $N = 1$, as shown in [11], the change of transmission does not exceed 0.01; therefore, the change of $j$ has no noticeable effect on the evolution of the transmission spectrum (see Figure 1b). We observe that already at $N = 100$ oscillations in the transmission occur at the red side edge of the Bragg resonance visible as the alternate yellow and red colors highlighting a regular oscillation of transmittivity form about 0.6 to about 0.8. Increasing the intensity of the oscillations corresponds to higher modulation of the transmittivity (alternate green and red colors) and occurs for a wider range of wavelengths. They also become evident in the blue edge of the Bragg resonance, even if showing a weaker modulation and a narrower range of wavelengths. At $N = 400$, we observe a remarkable narrowing of the PBG (deep blue stripe), which corresponds to irregular variable transmission on the red side edge of it. Figure 2c shows the spectra for further increase in the intensity from $N = 500$ to $N = 5000$. We also observe a broadening of the wavelength range of instabilities occurring in the blue side edge of the spectrum. A large increase in the intensity shows the loss of any regular behavior with the disappearance of the PBG.

In order to better highlight the feature of the observed phenomenon, we plot the transmission vs. $j$ for two wavelengths $\lambda_1 = 494.4$ nm and $\lambda_2 = 510.4$ nm symmetric with respect to the Bragg peak $\lambda_B = 502.4$ nm on the blue side and on the red side of the edge of the resonance. These curves are reported in Figure 3 for increasing values of the normalized intensity $N$. We observe that at $N = 150$, a stable value of the transmitted intensity is quickly achieved for both wavelengths (Figure 3a). At $N = 500$, a stable value is reached after a few damped oscillations at $\lambda_1$ while stable oscillations are present at $\lambda_2$ (Figure 3b).

A further increase in the intensity ($N = 650$) also gives rise to stable oscillations (usually of lower amplitude) at $\lambda_1$ when at $\lambda_2$ one already observes irregular oscillations (chaotic behavior) (Figure 3c) that are present at both wavelengths for higher intensities (Figure 3d, corresponding to $N = 700$).

We also pointed out the strong difference between the oscillatory behavior and the unstable variable values shown at high intensities by calculating the Fourier transforms of these curves. We report in Figure 4 two examples related to the behavior shown at $\lambda_2 = 510.4$ nm for $N = 500$ and $N = 700$. In these plots, we see a clear peak in the FT spectrum, showing that oscillation is occurring with a narrow frequency range for the first case, while at higher intensities no clear oscillation frequency can be identified. This result highlights the occurrence of two different regimes in the nonlinear propagation of the light at this wavelength.

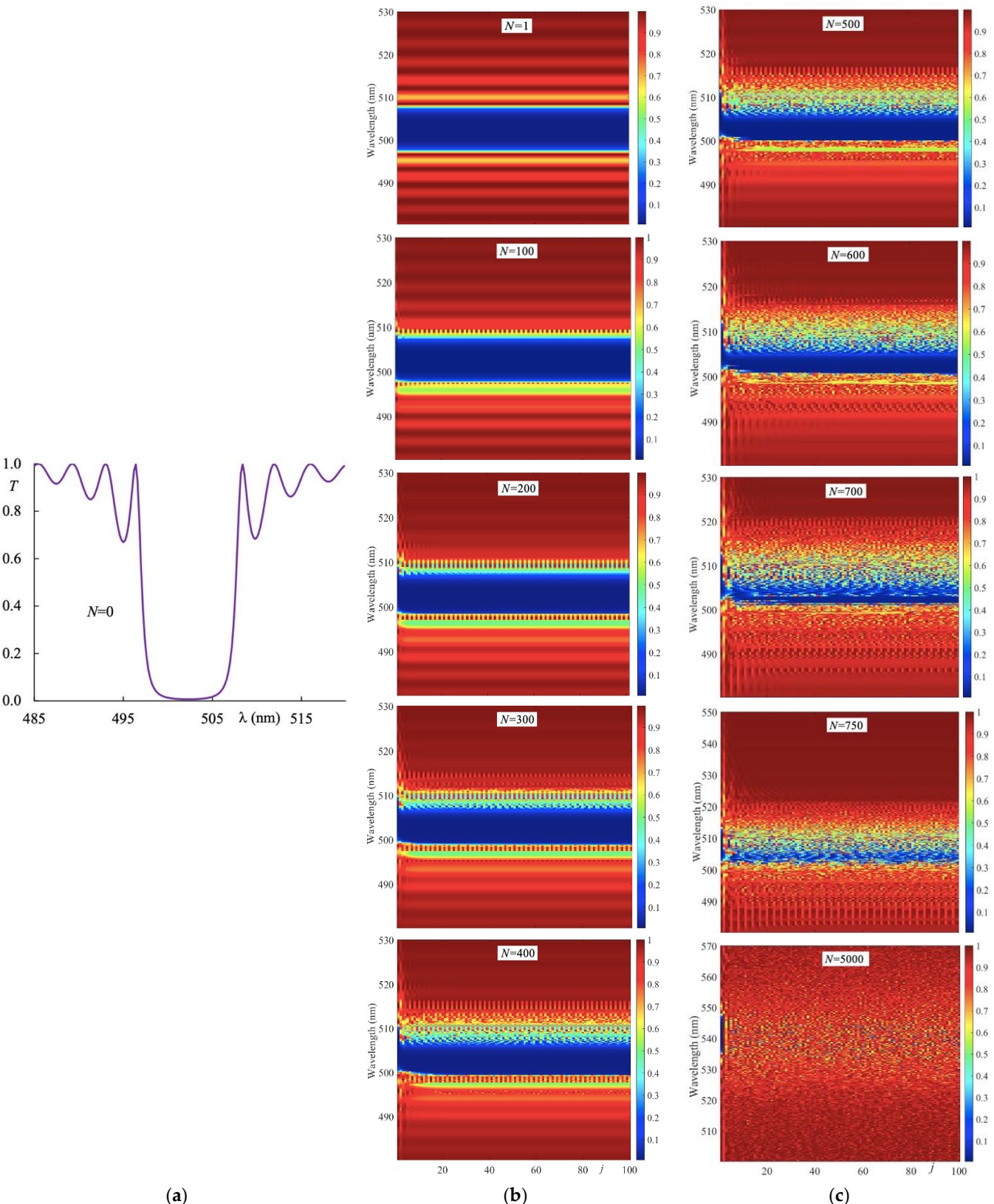

**Figure 2.** The transmission spectra vs. the step index *j* for increasing values of the normalized intensity. Colors give the normalized value of the transmittivity. The index *j* is proportional to the elapsed time (see text). (**a**): spectrum at *N* = 0 reported for *j* = 1 since it does not change with *j*; (**b**) *N* = 1 up to *N* = 400, where instabilities begin on the red side edge of the PBG; (**c**): *N* = 500 up to *N* = 5000, where the PBG disappears.

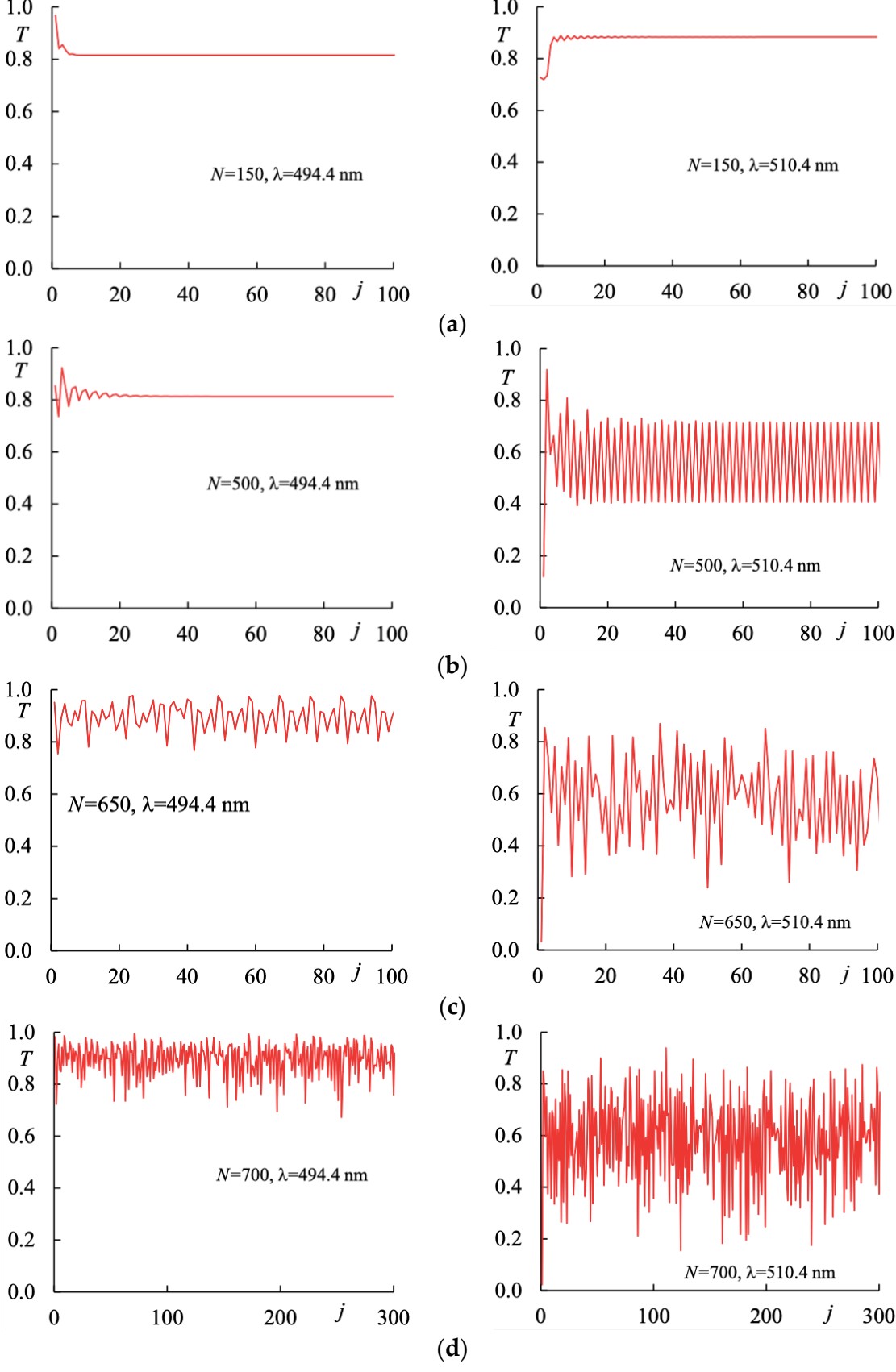

**Figure 3.** The transmittivity vs. the index *j* for two wavelengths $\lambda_1$ = 494.4 nm and $\lambda_2$ = 510.4 nm symmetric with respect to the Bragg peak $\lambda_B$ = 502.4 nm at increasing values of the normalized intensity *N*. (**a**) *N* = 150; (**b**) *N* = 500; (**c**) *N* = 650; (**d**) *N* = 700.

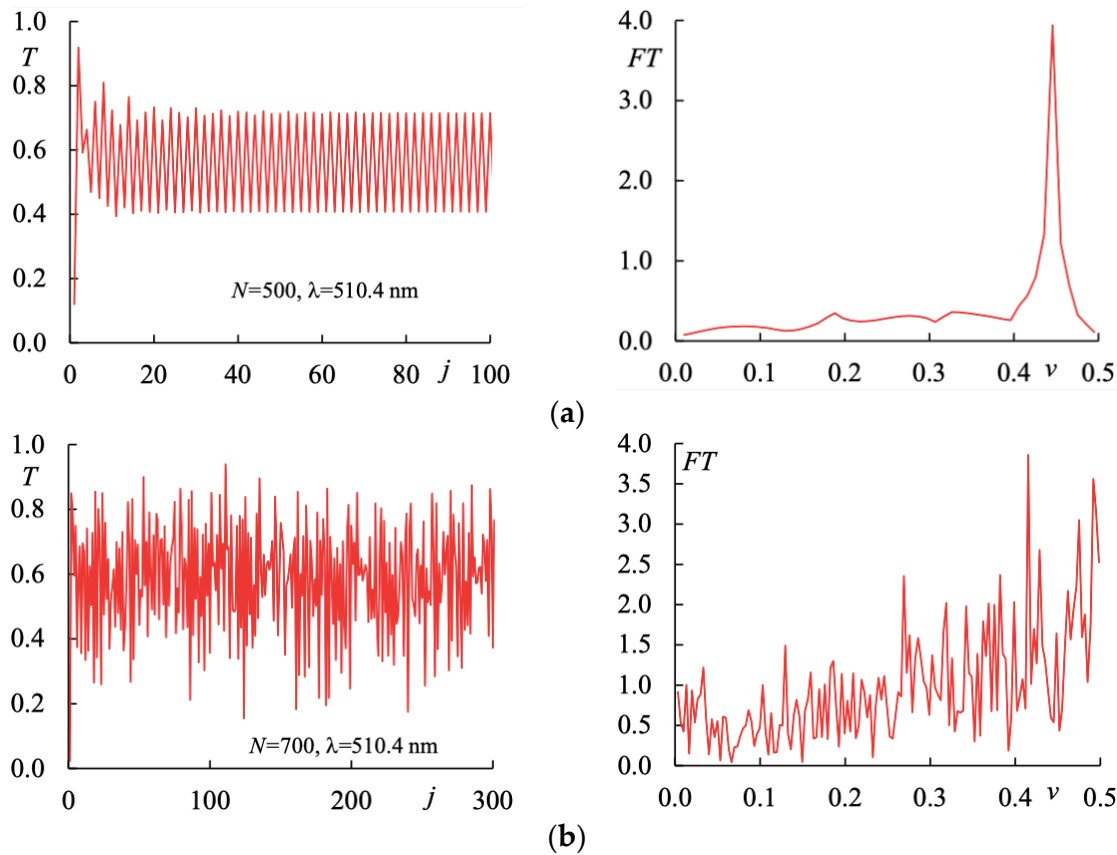

**Figure 4.** The transmittivity vs. the index $j$ (left curve) and the corresponding Fourier transform (FT) (right curve) on the red side edge of the Bragg resonance at $\lambda_2 = 510.4$ nm. (**a**): $N = 500$, leading to stable oscillations with a narrow range of frequencies shown by FT; (**b**): $N = 750$, leading to unstable values for the transmission confirmed by FT.

## 3. Discussion

The amazing self-induced oscillatory behavior occurs at intensities easily achievable by a c.w. laser with moderate power, in fact, $N = 500$ (just above the threshold intensity for $\lambda_2 = 510.4$ nm) corresponds to $I = 1.9 \cdot 10^5$ W/cm$^2$. A simple model allows an understanding of the onset of the oscillations; it is illustrated in Figure 5 where the transmission spectrum at $j = 1$ is plotted. It has already been demonstrated that optical reorientation leads to a redshift of the Bragg resonance, as summarized by the above-reported Equations (5) and (6) fulfilled by the experimental data reported in [7,8]. Then, by considering a narrow range of wavelengths on the red side edge of the PGB (indicated by the red arrow in the figure) we notice that for these wavelengths the redshift of the Bragg resonance means a decrease in transmittivity (red curve). This results in a decrease in light intensity. The consequence is an increase in the effective static field $E_{eff}$ with consequent blueshift of the resonance leading in turn to an increase in transmittivity and light intensity inducing again a redshift of the resonance able of restarting the process (blue curve). This model also explains the strong asymmetry reported in Figure 3, which leads to oscillatory behavior at higher intensities for wavelengths on the blue side edge of the PBG. In fact, in this case, the same process occurs around the secondary maxima of the resonant spectrum and not next to the PBG; thus the oscillatory behavior is observed at a higher intensity with a lower amplitude of modulation.

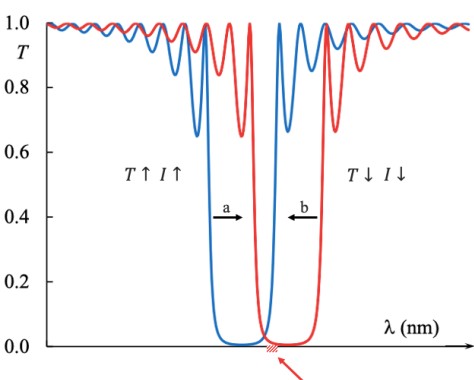

**Figure 5.** The transmission spectrum at $j = 1$ undergoing alternate redshift (a) and blueshift (b) following the change of intensity of the light beam due to the consequent change in transmission for the range of wavelengths indicated by the red arrow (see text).

It is quite important that at the fixed impinging intensity the effect expected at a given wavelength it is determined by its distance $\delta\lambda$ from $\lambda_B$ and occurs at a lower intensity on the red side edge of the resonance. As a matter of fact, $\delta\lambda$ depends on the static field $E_S$ applied to the ChOH sample, which determines the location of $\lambda_B$ on the wavelength axis. This fact offers the opportunity of using the static field $E_S$ as a parameter to drive the transmittivity behavior for a light beam at a specific wavelength. It means that the optical transmission at a fixed wavelength and light intensity can present a stable value, a self-oscillatory behavior, or a chaotic one depending on the applied electric field.

As an example, we show in Figure 6 the calculated transmission vs. $j$ at $\lambda_1 = 494.4$ nm at different intensities for two different values of the applied static field. We observe (Figure 5a) that with $N = 150$ we get a stable value for the light transmission with an applied field of $E_S = 2.03$ V/mm, while at $E_S = 2.09$ V/mm we observe the onset of stable oscillations. Alternatively, at $N = 650$ (Figure 5b) we have stable oscillations at $E_S = 2.03$ V/mm and chaotic behavior with $E_S = 2.09$ V/mm. This result shows that we have a system that can be driven to oscillatory or chaotic behavior by a static electric field and light intensity that is able to switch the system between different states. These properties need further theoretical and experimental investigation.

It is important to remark that all data reported in Figures 2–4 show the actual time dependence of the transmitted intensity because the step index $j$ can be considered as the elapsed time in units of the response time $\tau_{on}$ required by the material to adapt its parameters to the new field values: $j \sim t/\tau_{on}$. In this way, we are also able to foresee the time range of the observed oscillations. In fact, they are related to small changes in the tilt angle $\theta$ determining the bend distortion of the helical structure. The maximum value of $\tau_{on}$ is the relaxation time $\tau_{off}$ necessary for the system to come back to equilibrium after switching off the exciting optical field, and in general, is shorter and shorter for increasing intensities. Therefore, we can evaluate the order of magnitude of the response time from the expression used when considering a bend distortion in homeotropic nematics [14]:

$$\tau_{off} = \frac{\gamma_1}{K_3}\left(\frac{d}{\pi}\right)^2$$

and find that with typical values for the bend elastic constant $K_3$ and the viscosity $\gamma_1$ for a sample thickness of 20 μm gives $\tau_{off} \sim 40$ *ms*. However, we must take into account that ChOH exists only when $K_3 < K_2$, in fact, its value has been recently measured [15] to be 2–3 times lower than in nematics; therefore it is possible a longer response if a similar value of the viscosity is used. For this reason, we can expect these oscillations to occur in the tenth of a millisecond range. A more detailed analysis is underway on this feature to compare the variation of the oscillation frequency at different intensities and experimental data are necessary to have a deeper insight into this matter.

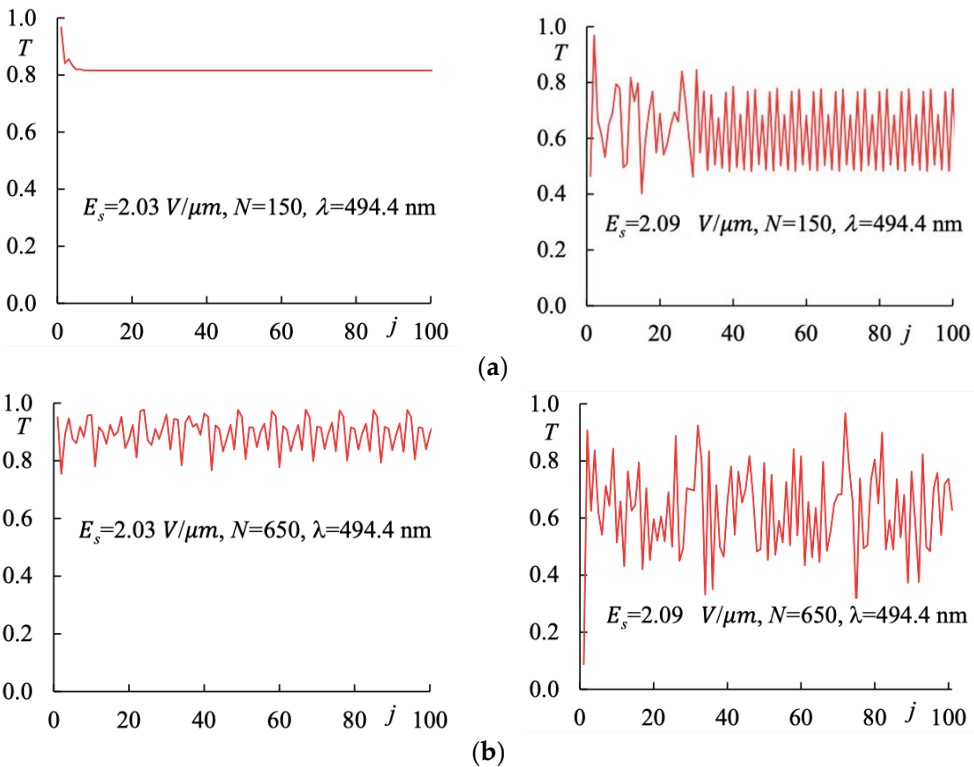

**Figure 6.** The transmittivity vs. the index $j$ at wavelengths $\lambda_1 = 494.4$ nm for two different values of the applied static field: $E_S = 2.03$ V/mm (curves on the left) and $E_S = 2.09$ V/mm (curves on the right): (**a**) $N = 150$; (**b**) $N = 650$.

## 4. Conclusions

Here we have reported a detailed study of the nonlinear light propagation of a beam traveling along the helix direction of a heliconical cholesteric liquid crystal, using Ambartsumian's layer addition modified method. The nonlinear light propagation is considered an effect of the optical reorientation induced by the light beam on the molecular director and the time evolution of the whole transmission spectrum is analyzed at increasing values of light intensity. We have reported how this spectrum is modified by increasing the light intensity, showing the onset of damped oscillations first followed by stable oscillations and by irregular (chaotic) behavior at higher intensities. We have demonstrated the strong asymmetry of the self-induced effects. In fact, they take place at lower intensities on the red side edge of the Bragg resonance while stable values of transmittivity are still observed on the blue side edge. A further increase in the intensity leads to oscillations of lower amplitude on the blue side while an irregular behavior of the transmission is achieved on the red side. At higher intensities, the Bragg resonance disappears and transmission becomes unstable for any light wavelength. A simple phenomenological model is proposed that explains both the onset of the oscillatory behavior and the wavelength asymmetry on the opposite sides of the Bragg resonance. In fact, since optical reorientation leads to the redshift of the Bragg peak, for wavelengths near the long-wavelength boundary of the PGB the redshift means a decrease in transmittance. This leads to a decrease in the intensity of the light. A consequence is an increase in the effective static field $E_{eff}$ acting on the structure followed by a blueshift in resonance, which in turn leads to an increase in transmittance and light intensity, again causing a redshift of the resonance, which can restart the process. This model also explains the strong asymmetry, which leads to oscillatory behavior at higher intensities for wavelengths at the blue edge of the resonance. In fact, here the same process occurs around the secondary maxima of the resonance spectrum, so the oscillatory behavior occurs at higher intensities with lower amplitude.

We also point out that the static electric field is a driving parameter to switch from stable to oscillatory to irregular behavior in the transmittivity at a given light wavelength.

It is remarkable that we have a system with tunable "chaotic" behavior driven by light intensity and external static electric field that has already been shown in a previous work [11] to correspond to a disorder of the conical structure changing both the conical angle and pitch through the sample thickness. This is potentially interesting for application development as it has been underlined in Ref. [16] where it has been shown that a completely disordered and scattering optical medium can sometimes outperform the most carefully designed and engineered photonic device.

Finally, it is important to underline the difference between the effect of the light-induced shift of the Bragg resonance discussed in this paper as well in the previous papers related to an optical reorientation in ChOH and the changes in the resonance obtained in conventional CLC, reported in a number of articles in the past [17–19]. Here we have the direct interaction between the optical field and the director leading to reorientation and consequent change of the helix pitch, while the changes of the Bragg resonance observed in conventional cholesterics were obtained through trans-cis isomerization, leading to a conformational change of the molecules.

**Author Contributions:** Conceptualization A.H.G. and F.S.; theoretical approach F.S.; numerical calculations A.H.G.; data analysis A.H.G. and F.S.; writing—review and editing, A.H.G. and F.S. All authors have read and agreed to the published version of the manuscript.

**Funding:** This research received no external funding.

**Institutional Review Board Statement:** Not Applicable.

**Informed Consent Statement:** Not Applicable.

**Data Availability Statement:** Data are available from A.H.G.

**Acknowledgments:** A.H.G. acknowledges the Foundation for the Advancement of Theoretical Physics and Mathematics "BASIS" (Grant № 21-1-1-6-1).

**Conflicts of Interest:** The authors declare no conflict of interest.

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
