# Peer review of "Light-Induced Self-Oscillations and Spoiling of the Bragg Resonance Due to Nonlinear Optical Propagation in Heliconical Cholesteric Liquid Crystals"

_photonics, doi:10.3390/photonics9110881_

Round 1

Reviewer 1 Report

An interesting paper summarizing the results of a thorough study of nonlinear optical processes in relatively new class of materials, heliconical cholesteric liquid crystals. It deserves publication as is, however, the paper would benefit from the minor changes discussed below.

First, the abstract claims “self-induced oscillations take place at lower intensities on the red side edge of the Bragg resonance” which is probably correct for a given material under given conditions that need be indicated.

The phenomenon as described by the authors, “originated by the light-induced shift of the Bragg resonance that affects light transmission during propagation”. However, it is well known that light could induce shift both towards red as well as green wavelengths. Moreover, the process depends on geometry. It is probably assumed that light propagates along the helix axis. An additional figure showing the modelled conditions may help.

A clear summary of the difference in processes in heliconical CLCs compared to processes in conventional CLCs could be very helpful too for the reader given strong interactions of light beams with conventional cholesteric LCs causing dramatic changes in bandgap have been studied in a number of papers, for example, cited below:

Jonathan P. Vernon, et al., Photostimulated control of laser transmission through photoresponsive cholesteric liquid crystals, Opt. Express 21(2), 1645-1655, 2013.

U.A. Hrozhyk, et al., Nonlinear optical properties of fast, photoswitchable cholesteric liquid crystal bandgaps, Optical Materials Express, 1 (5), 943-952, 2011.

S. V. Serak, N.V. Tabiryan, T. J. Bunning, Nonlinear transmission of photosensitive cholesteric liquid crystals due to spectral bandwidth auto-tuning or restoration, J. Nonlinear Optical Physics & Materials, 16 (4), 471-483, 2007

Author Response

Rev#1

  • First, the abstract claims “self-induced oscillations take place at lower intensities on the red side edge of the Bragg resonance” which is probably correct for a given material under given conditions that need be indicated
  • The phenomenon as described by the authors, “originated by the light-induced shift of the Bragg resonance that affects light transmission during propagation”. However, it is well known that light could induce shift both towards red as well as green wavelengths. Moreover, the process depends on geometry. It is probably assumed that light propagates along the helix axis.

In line 2 of the abstract, it is specified the considered geometry “…light propagation of a beam traveling along the helix direction of a heliconical cholesteric liquid crystal showing that optical reorientation leads to instabilities in the optical transmission…”

Therefore, both geometry and the considered phenomenon are highlighted. Under these conditions in ChOH by increasing the light intensity only redshift of the Bragg resonance is possible because the optical torque increase the conical angle and the pitch increases as well. This is true for all the ChOH materials known up to date.

  • An additional figure showing the modelled conditions may help.

We thought it not necessary because we clarified several times in the text that we are considering propagation along the helix of the conical structure, therefore the optical field is circularly polarized in a plane orthogonal to it and all the referred studies are made in the same geometry (ref[1-3,6-8, 11]), however we included a new figure as suggested by the referee (Fig.1 of revised version)

  • A clear summary of the difference in processes in heliconical CLCs compared to processes in conventional CLCs could be very helpful too for the reader given strong interactions of light beams with conventional cholesteric LCs causing dramatic changes in bandgap have been studied in a number of papers, for example, cited below:…

Concerning optical reorientation, we have already pointed out in ref.7 that it is not very effective in conventional cholesteric liquid crystal since the twist reorientation is involved so that no effect on the helix pitch has been ever observed. Concerning the examples cited reported by the reviewer they refer to a photochemical process related to tran-cis isomerization of the molecules and not to what usually defined as optical reorientation involving a torque of the optical field on the molecular director. We have added a sentence in the conclusions to clarify this difference including the suggested references.

Reviewer 2 Report

I think the work presented on the study of the nonlinear light propagation of a beam traveling along the helix direction of a heliconical cholesteric liquid crystal, with optical reorientation and instabilities in the optical transmission when the light wavelength is close to the Bragg resonance is worthy of publication. I think the work is unique and should be of interests to the readers in this area. 

I would recommend if the authors could include some schematics of the optical processes that they described in the content so that they provide a better visual aid to the readers. In addition, can the authors comment on the deviation of the computational results presented due to variations from the physical dimensions of the actual structures: observed in experiments? Also, how does the model account for these deviations?

Author Response

  • I would recommend if the authors could include some schematics of the optical processes that they described in the content so that they provide a better visual aid to the readers
  • We thought it not necessary because we clarified several times in the text that we are considering propagation along the helix of the conical structure, therefore the optical field is circularly polarized in a plane orthogonal to it and all the referred studies are made in the same geometry (ref[1-3,6-8, 11]), however we included a new figure as suggested by the referee (Fig.1 of revised version)

  • In addition, can the authors comment on the deviation of the computational results presented due to variations from the physical dimensions of the actual structures: observed in experiments? Also, how does the model account for these deviations?

We do not understand the question because the experiments made up to now concern a pump-probe configuration where a beam (pump) out of the Bragg resonance induces the director reorientation that is tested by a probe beam (white source) of very low intensity. On the contrary, in the present paper we consider a single beam and the effect is self-induced. The thickness is always about 20 mm.

Reviewer 3 Report

The manuscript titled “Light-induced self-oscillations and spoiling of the Bragg resonance due to nonlinear optical propagation in heliconical cholesteric liquid crystals” by Ashot H. Gevorgyan and Francesco Simoni reported a detailed study about the phenomena and the mechanism of light-induced oscillations and destroy of the Bragg resonance. The whole transmission is modified differently at lower light intensity, further increased intensity, and higher intensity. It is a topic of interest to the researchers in the related areas but the paper still needs improvement before acceptance for publication. My detailed comments are as follows:

1.         It will be valuable for comparisons to be made between heliconical CLCs (in this study) and traditional CLCs (Photonics Res. 2022, 10, 786-792.) in introduction and discussions, which would provide a more comprehensive picture for the audience. Why only the heliconical CLC system exhibits such phenomena.

2.         In the abstract, the authors mentioned a recent paper in the first sentence. This should be made explicitly clear. I think it is ref.[9].

3.         What is the significance of studying the self-oscillations and spoiling of the Bragg resonance? What can it be used for and why it is important?

4.         In Page 2, the author mentioned a pump-probe configuration. How do the pump beam and probe beam work?

5.         The effects of such optical reorientation on the nonlinear propagation of a light beam inducing optical reorientation has also been theoretically investigated in ref.9. The author should focus more on the new significant findings.

6.         N is 700 (fig.3b) or 750 (Line 299)? The Figures in Figs.2b/d are the same as Figs.3a/b(left). I recommend that Figure 2 and 3 should be merged for clarity.

7.         The authors mentioned Ref.16 in the conclusion. They should explicit the relationship between the “chaotic” behavior in this study and the disorder in ref.16. How to utilize the tunable “chaotic” behavior?

Author Response

  • It will be valuable for comparisons to be made between heliconical CLCs (in this study) and traditional CLCs (Photonics Res. 2022, 10, 786-792.) in introduction and discussions, which would provide a more comprehensive picture for the audience. Why only the heliconical CLC system exhibits such phenomena?

The difference between Heliconical Cholesterics and traditional CLCs has been described in several previous papers (ref.[1-3,7] and the reason why only ChOHs exhibit this phenomenon has been underlined in ref.[ 7], however we have included a couple of sentences in the introduction to remind it as suggested by the referee.

  • In the abstract, the authors mentioned a recent paper in the first sentence. This should be made explicitly clear. I think it is ref.[9].

We don’t think references should be indicated in the abstract, this is done in the text. The paper is ref.[11]

  • What is the significance of studying the self-oscillations and spoiling of the Bragg resonance? What can it be used for and why it is important?

The study of nonlinear systems producing self-oscillations and different routes to chaos has been a research subject since more than four decades. Most of them are artificial systems designed for that, under some conditions liquid crystals are among the few cases that spontaneously show this behavior (see for instance Phys.Rev.E 47, 1741 (1993) and references therein) therefore the present case looks as a new original condition where this behavior is achieved. From point of view of applications these processes can be of interest in the field of optical communications and optical computing.

  • In Page 2, the author mentioned a pump-probe configuration. How do the pump beam and probe beam work?

The pump-probe configuration is described in ref.[7]: one beam (pump) out of the Bragg resonance induces the director reorientation that is tested by a probe beam (white source) of very low intensity.

  • The effects of such optical reorientation on the nonlinear propagation of a light beam inducing optical reorientation have also been theoretically investigated in ref.9. The author should focus more on the new significant findings

We actually do not understand the suggestion since the new findings are the subject of all the paper: the behavior in the whole spectrum around the Bragg resonance, the strong asymmetry, the switching from one regime to a different one by changing the applied static field, etc.

  • N is 700 (fig.3b) or 750 (Line299)? The Figures in Figs.2b/d are the same as Figs3a/b(left). I recommend that Figure 2 and 3 should be merged for clarity

Thank you for the note, we have corrected the wrong value in the text as 700. We believe the presentation is better leaving Figure 2 and 3 as separate figures.

  • The authors mentioned Ref.16 in the conclusion. They should explicit the relationship between the “chaotic” behavior in this study and the disorder in ref.16. How to utilize the tunable “chaotic” behavior?

Actually, the word chaotic is used for the transmission output, we have shown in a previous paper (ref.[11]) that it is associated to induced disorder of the helical structure. Tuning the optical disorder can be useful for random lasers. We added a sentence in the conclusions.